# Impact of Digital Platform Organization on Reducing Green Production Risk to Tackle COVID-19: Evidence from Farmers in Jiangsu China

Lishi Mao, Junfeng Song, Siyuan Xu and Degui Yu *

Digital Rural Research Institute, Nanjing Agricultural University, Nanjing 210095, China
* Correspondence: yudgu@njau.edu.cn

**Abstract:** The agricultural organization based on digital platforms in C2F (Company–Platform–Farmers) may be an effective way to reduce the increased risk of green production caused by the COVID-19 pandemic, but the specific mechanism and impact involved are unclear. Applying risk cognition and decision theory, we built the theoretical framework on platform organization, pandemic risk, perception value, and green production continuity, and ascertained the impact effects and path using the PLS-SEM method. We found that the risk of COVID-19 overflow had a significant impact on farmers' green production continuity by mediating perception value, especially regarding reduced green technology adoption intention and increased cost of living. Utilizing perfect platform value cognition, participation co-operation, access and benefit distribution, and safeguard and restraint measures of platform organization in C2F, we offered a new approach to reduce the green production risks caused by COVID-19, such as material and labor shortages, financial pressure, sales channel blockages, and price volatility. We explained the behavior-moderating motivation of farmers with different risk preferences and subjective norms in relation to tackling COVID-19. We should aim to perfect the functions of digital platform organizations to optimize the benefit linkages in C2F, accelerate farmers' digitization ability cultivation to increase the cognitive risk level, and strengthen the policy guidance of COVID-19 prevention to reduce the influence of farmers' subjective norms.

**Keywords:** COVID-19; digital platform; benefits-linking stability; green production risk; C2F model



## 1. Introduction

The continuous recurrence and prevention tactics used in response to the COVID-19 pandemic have increased the risk of family farms' green production. Developing green agriculture is at the heart of the Rural Revitalization Strategy in China, and the family farm is the focus of green agriculture development. According to the third agricultural census in 2017, with an annual growth rate of 26.5% and an average farm scale of 11.7 ha. A total of 54.9 thousand family farms are currently operational, occupying 461.1 million ha of cultivated land. COVID-19 prevention measures, such as traffic control and forced quarantine, presented an unprecedented challenge, and increased the risk for farmers [1]. The public was required to quarantine at home [2], which created a labor shortage. Due to the deepening of the social divisions of the labor system, the supply of pre-production in agriculture, the availability of socialized services during production, and the sale of post-production agricultural products all depend on the market, and traffic control [3] leads to serious production obstacles, interruption of the logistics chain of the market, and delays in the farming season [4]. Many dealers and agents were unable to buy and sell products, leading to an imbalance of supply and demand in the market, as well as pricing instability [5]. The characteristics of agricultural production, such as a long investment return period and strong market dependence, determine that agricultural risks such as market, nature, technology, and policy are a greater threat, which increases the vulnerability, exposure, and sensitivity of small-scale farmers' decentralized production

processes [1]. This indicates that green agriculture faced greater risks and uncertainties during the pandemic. The use of chemical fertilizers and pesticides was reduced, resulting in higher technical requirements and higher input costs. Therefore, it is important to explore the impact of pandemic risk on continuous green production and construct a set of effective risk-response measures to maintain farmers' livelihoods while continuing to tackle COVID-19.

The new model of agricultural organization based on a digital platform has provided an important means to reduce the risk overflow from COVID-19. Chinese agriculture has formed a digital platform-based internet business model called C2F (Company–Platform–Farmers), which was conducive to improving green agriculture development because of platform features such as digitization, marketization, and organization. Using last-generation digital technologies, such as the Internet of Things (IoT) and Artificial Intelligence (AI), the digital platform of C2F provided a new networked Farm Management Information System (FMIS) that could ensure output and reduce the use of water, fertilizer, and pesticides. Based on a shared economy, the C2F carries out joint production and management. It is characterized by organized open distribution, and it integrates a variety of resource allocation mechanisms [6], making it an important measure to revitalize inventory and improve efficiency and service provision [7]. Through the benefit-linking of the digital platform [8], farmers, enterprises, platforms, and governments can be associated with management and supervision [9], rationally arrange green production plans, optimize resource allocation [10], and improve farms' digital level [11]. By realizing product digitization to increase profits, market branding to reduce costs, and operation platform to reduce risks, C2F provides opportunities to prevent the risk overflow of COVID-19.

We conducted extensive research on the impact of organization on farmers' green production based on risk perception and decision theory. Policy regulation, benefit incentive, and risk cognition exerted a main effect on people's decisions [12]. Driven by benefit incentives, the organization improved farmers' social capital to adapt green production through regulatory constraints, value co-production, and value sharing [13], with a lasting impact on the prioritization of choices [14]. As a social capital, agricultural organizations can actively benefit incentives through value sharing, which in turn, has a positive impact on their green production [15]. The policy regulation of an organization included normative factors such as trust, fairness, and reciprocity [16,17], which promote social and cultural identity [18–20], while subjective norms ensured the coordination and sustainable behaviors of farming groups. Through unified management and service of organizations, a closer benefits linkage could effectively achieve the effect of "risk sharing and benefit sharing" [21], realize different farmers' balance benefits [8], and improve the overall ability to withstand risks. The level of farmers' risk cognition (risk perception and risk preference) determined their decisions [22], because most farmers relied on intuitive risk judgment for green production [22–24]. With the increased risk of COVID-19 overflow, the impact of the effects and path of digital platform organization in C2F on continuous green production requires additional research.

Based on the C2F, we developed a green product for family farmers, regional farmers (users) were taken as the research subjects. Applying risk decision theory, we built a theoretical framework based on platform organization, pandemic risk, perception value, risk preference, subjective norms, and production continuity, and ascertained the impacted effects and path using the PLS-SEM model. We analyzed the impact of the risk of COVID-19 overflow on production continuity, demonstrated the possibility of digital platform organization to withstand pandemic risk, studied the behavior moderating motivation of farmers with different risk preferences and subjective norms in tackling the pandemic, and provided suggestions or measures for maintaining farmers' livelihoods while continuing to tackle COVID-19.

## 2. Research and Theoretical Framework

### 2.1. Benefits-Linking Stability on Platform Organization in C2F

In the C2F (Company–Platform–Farmers) model, as a new type of farmer cooperative organization uses the leading company or enterprises as the core and the cloud farm platform as the means, the digital platform achieved enterprise-driven industrialization for green agriculture [8]. The central feature of the digital platform is that benefits-linking stability could effectively reduce farmers' green production or market risk.

The digital platform reshaped digital green agriculture organization. From agricultural material, farm sites, and production to sales, the whole management and information transmission process has changed from its original chain-like linear or island-like configuration, to the current circular and network-like basic sharing transmission mode [25]. The C2F created the future distributed and platform-based resource sharing and transmission between regional farms, in which each management link can interact and share information with other links in real-world time.

Benefits-linking stability (PS) refers to the stability and compactness of the digital green industrial consortium based on platform organization in C2F, including platform value cognition (PS1), participation value co-operation (PS2), access and benefit distribution (PS3), and safeguard and restraint measures (PS4).

### 2.2. Research Theoretical Model

According to the digital platform features and business of C2F, using risk cognition and decision theory, we established a research model to measure the green production continuity to tackle the risk of COVID-19 overflow (in Figure 1).

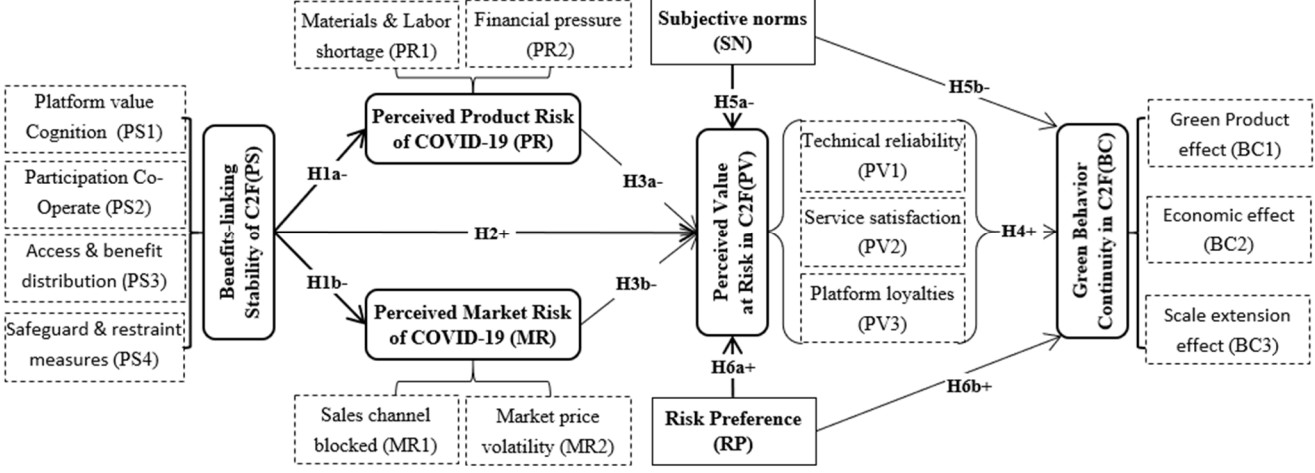

**Figure 1.** Research theoretical model.

Due to the impact of COVID-19, the perceived product risk (PR) was related to the uncertainty and severity of materials, labor shortages (PR1) and financial pressures (PR2), and the perceived market risk (MR) was measured by blocked sales channels (MR1) and market price volatility (MR2). The perceived value at risk (PV) was based on the overall satisfaction of the platform organization in C2F in relation to the technical reliability (PV1), service satisfaction (PV2) and platform loyalties (PV3). Green production continuity (BC) refers to the improvement in farmers' livelihoods in terms of creating green products used to tackle COVID-19, including the green product effect (BC2), economic effect, (BC1) and the scale extension effect (BC3).

In addition, two moderating variables were introduced to this research model to regulate the effect of PV on BC, such as the subjective norm (SN), including the community impact and government influence of C2F, and the risk preference (RP), measured as risk averse (RP1), risk neutral (RP2), or risk appetent (RP3).

### 2.3. Research Hypothesis

**Hypotheses 1 (H1):** The benefits-linking stability of a platform organization in C2F (**H1a-**: perceived product risk; **H1b-**: perceived market risk) significantly reduced the risk overflow from the COVID-19 pandemic.

With closer and more stable benefit linkages, the agriculture organization based on the digital platform in C2F enhanced the family farms' capacity to resist short-term risks to green production, such as COVID-19. Based on this platform, the regional farm decentralized management organized 'six unifications' by the leading enterprises, such as unified brand, standard, material (fertilizer or pesticides), production, services, and market channels, and formed a new type of digital green industrial consortium and an effective data-sharing model to help farmers reduce the production and market risks by improving platform value cognition and participation value co-operation [26]. With the perfect access, benefit distribution, safeguarding, and restraint measures of the digital consortium, the farmers' cognitive ability with regard to risk aversion and perception would be effectively improved, reducing the uncertainty of their production [27,28].

**Hypotheses 2 (H2):** The benefits-linking stability of a platform organization in C2F had significant positive effects on the farmers' perceived value of the risk of COVID-19.

Perceived value is the subjective evaluation of risk based on people's own benefits [29,30], which obviously affects their behavior [31,32]. The magnitude of perceived value depends on the rational judgment of 'gain and loss'. It has also been proposed that when individuals perceive a greater personal benefit of goods or services, the level of value perception increases [33]. A stable benefit-linkage platform provides farmers with agricultural technology information and management services [34] so that farmers can improve their trust in technology and service satisfaction, and thus, improve their value perception.

**Hypotheses 3 (H3):** The perceived risk (**H3a-**: perceived product risk; **H3b-**: perceived market risk) had significant negative effects on the farmers' perceived value in relation to the risk of COVID-19.

Perceived risk refers to perceptions of the environment as having adverse effects on humans [35]. Risk and return have always been the same; high returns often carry a high risk. As a "rational economic man", farmers often weigh risk and value before making decisions. Individual risk perception has a significant impact on willingness and attitude [36,37]. In the case of COVID-19, farmers are facing production pressure due to a reduced workforce and sales being blocked due to uncertain factors such as pandemic prevention policies, which indicates that they have reduced trust in technology and service guarantees provided by platforms. The improvement in farmers' risk perception will decrease their value perception.

**Hypotheses 4 (H4):** The perceived value had significant positive effects on the farmers' green production continuity in relation to the risk of COVID-19.

Perceived value has a great influence on farmers' behavior [38–40]. As a kind of organization, the C2F platform provides technology and other services [41] to ensure that farmers can obtain the relevant knowledge and skills concerning the production and management process. Thus, farmers will have enough money and technology to produce green products. When farmers have a high perception of production value, they will improve their willingness to participate in green production, and even take the initiative to increase varieties, adopting the latest green technologies.

**Hypotheses 5 (H5):** The subjective norms had significant negative effects on farmers' perceived value (**H5a-**) and green production continuity (**H5b-**).

Subjective norms are the social and public opinion pressures felt by farmers when adopting green production, such as the supervision and constraints of social systems, moral

norms, and media public opinion in the surrounding environment, which will gradually restrict or guide farmers' behavioral norms [42]. Many pandemic prevention policies have put pressure on production and logistics; at the same time, the invisible societal norm of fighting the pandemic will make farmers stay at home as much as possible to reduce the spread of COVID-19 [43]. Therefore, when farmers are more affected by subjective norms, their perceived value will be reduced, and green production will also be reduced.

**Hypotheses 6 (H6):** The risk preference dimension had significant positive effects on farmers' perceived value (**H6a-**) and green production continuity (**H6b-**).

Risk preference plays an important role in farmers' decision-making [44]. Affected by the pandemic, consumers' pursuit of green consumption and healthy quality has increased [45]. Consumers are more willing to pay for the functional and social values of green agricultural products [46,47]. The impact of COVID-19 on green production has another feature. We graded the risk appetite from conservative to risk-taking. Conservative farmers-those with a low-risk appetite-will be averse to the market risks posed by COVID-19 and the production risks caused by quarantine policies. Farmers who love to take risks perceive more value in such an environment and pursue higher green agricultural profits during the COVID-19 pandemic, so they are more willing to take the initiative to obtain information and carry out green production.

### 3. Material and Methods

*3.1. Measurement Design*

This study aims to determine the impact path of benefits-linking stability of platform organization in C2F on farmers' green production behavior continuity while tackling the risk of COVID-19. Based on the research model in Figure 1, several factors may affect farmers' green production behavior, such as risk perception, risk value perception, subjective norms, and risk preference, etc. All the path construct variables were measured using multi-item perceptual scales and are rated on a five-point Likert scale, with scores ranging from 1-strongly disagree, 2-disagree, 3-uncertainty, 4-agree, and 5-strongly agree. A total of 16 latent variables and 31 item observed variables in perceptual scales are shown in Table A1.

*3.2. Data Sources*

We selected the Jiangsu Province as the research region because its agricultural development is representative of China. According to the statistics on the Jiangsu Government Website, the level of digital rural development and agriculture reached 65.4% in Jiangsu and is top ranked nationwide. Jiangsu has significant advantages regarding the use of the digital platform model. More than 0.1 million family farms have adopted the various cloud platforms (e.g., TaoBao, JingDong, and WeChat), and the provincial service platform has more than one million farmer users. There are more than 7700 leading companies involved in more than 1000 agricultural industrialization consortia, which has driven the development of 6.7 million farmers through various digital platforms; 19.6% of farmers are developing facility agriculture. The research obtained represents a good reference significance for internet agriculture in China.

The analytical data were derived from the family farmers questionnaire on the cloud farm platform (CFP V1.0) that we developed for family farms. The platform mainly provides green production and management, as well as technical and market services for fruit and vegetable production based on the C2F model. In total, 15,000 farmers with an area of 4000 ha have become involved with the CFP in the last 3 years. The questionnaire was designed according to Appendix A. A stratified random sampling method-probability-proportional-to-size sampling-was used in the survey. The investigated farmers (the person in charge of farm production and operation) in each of the three economic regions, southern, middle and northern Jiangsu, were chosen via random sampling. Additionally, five prefectural counties (cities and districts) were selected from each economic zone.

Then, stratified random sampling was used to select 20 family farms using the cloud farm platform from each county. Prior to each interview, we called the participant to explain the purpose and general content of the interview and make an appointment to conduct the interview, so the rejection rate of the survey subjects was low. If rejected, we returned to the users' database for additional sampling. The interviewee was the main person in charge of production and operations on the family farm. Two hundred and eighty-two questionnaires were assessed in the summer of 2021, of which two hundred and sixty-four questionnaires were valid, 93.62% in total, with some missing data and abnormal values. We used SPSS (V13.0) for data processing.

Table 1 presents a detailed view of survey farmers' demographics, including their sex, age, education, and farm scale. Among the surveyed farmers, 58.71% were males, 31.80% were part-time farmers, 41.67% were under the age of 50 (young and middle aged), and 30.68% had more than 12 years of schooling (high school education) with higher education. Most farmers' farms were about 50 mu in size (1 ha = 15 mu) with moderate-scale management. This demonstrated that these users played an important role in the platform organization, because of their higher cognitive and digital ability.

**Table 1.** Survey farmers demographics (*N* = 264).

| Measure | Items | Frequency | Percentage (%) |
|---|---|---|---|
| Sex | Male | 155 | 58.71 |
| | Female | 109 | 41.29 |
| Age (years) | <30 | 10 | 3.79 |
| | 31–50 | 100 | 37.88 |
| | 51–60 | 103 | 39.02 |
| | >60 | 51 | 19.32 |
| Education (years) [a] | ≤6 | 81 | 30.68 |
| | 9 | 102 | 38.64 |
| | 12 | 53 | 20.08 |
| | ≥14 | 28 | 10.60 |
| Farm scale (mu) [b] | 0–10 | 22 | 8.33 |
| | 11–50 | 121 | 45.84 |
| | 50–100 | 74 | 28.03 |
| | >100 | 47 | 17.80 |

Note: [a] years of schooling; [b] is unit of measure 1 ha = 15 mu. Source: Survey results, July 2021.

### 3.3. Reliability and Validity

We used SPSS (V13.0) to carry out the tests. The KMO value of the survey data was 0.942 > 0.9, and was based on Bartlett's sphere test, $\chi^2$ was 3305.49, while the degree of freedom (*df*) was 171 and $P < 0.01$, indicating that there was a significant correlation among the construct variables, which requires further analysis, as shown in Table 2.

**Table 2.** Test results using KMO and Bartlett.

| Project | | Test Value |
|---|---|---|
| Kaiser–Meyer–Olkin Measure of Sampling Adequacy | | 0.942 |
| Bartlett's sphericity test | Approximate chi-square | 3305.490 |
| | Degree of freedom (df) | 171 |
| | Significant (Sig.) | 0.000 |

In Table 3, the values on the main diagonal are the square root of the AVE of each latent variable, and the remaining values are the correlation coefficients between two latent variables, which need to be analyzed by Pearson correlation. For each dimension, the root value of the AVEs of all dimensions is greater than the correlation coefficient between dimensions, indicating that each dimension has good discriminant validity.

**Table 3.** Results of discrimination validity test of the scale.

|  | BC | MR | PR | PS | PV | RP | SN |
|---|---|---|---|---|---|---|---|
| BC | 0.821 | | | | | | |
| MR | −0.666 | 0.879 | | | | | |
| PR | −0.661 | 0.721 | 0.895 | | | | |
| PS | 0.532 | −0.597 | −0.681 | 0.805 | | | |
| PV | 0.810 | −0.737 | −0.826 | 0.700 | 0.922 | | |
| RP | 0.660 | −0.582 | −0.676 | 0.599 | 0.777 | 0.805 | |
| SN | −0.657 | 0.526 | 0.597 | −0.460 | −0.602 | −0.530 | 0.851 |

As shown in Table 4, the rotated eigenvalues of the first four principal factors of the nineteen indexes selected in this paper were 9.733 and 1.298, all of which exceeded 1, indicating that four principal factors could be extracted. The cumulative contribution rate was 58.059%, indicating that the four principal factors could reflect 58.059% of the information of the original data.

**Table 4.** Total variance interpretation.

| Component | Initial Eigenvalue | | | Rotated Square and Loading | | |
|---|---|---|---|---|---|---|
|  | Total | Percentage of Variance Cumulative | Accumulation% | Total | Percentage of Variance Cumulative | Accumulation% |
| 1 | 9.733 | 51.228 | 51.228 | 9.733 | 51.228 | 51.228 |
| 2 | 1.298 | 6.831 | 58.059 | 1.298 | 6.831 | 58.059 |
| 3 | 0.936 | 4.925 | 62.984 | | | |
| 4 | 0.831 | 4.373 | 67.357 | | | |
| 5 | 0.786 | 4.137 | 71.493 | | | |
| 6 | 0.683 | 3.597 | 75.091 | | | |
| 7 | 0.615 | 3.236 | 78.327 | | | |
| 8 | 0.549 | 2.890 | 81.217 | | | |
| 9 | 0.517 | 2.724 | 83.941 | | | |
| 10 | 0.505 | 2.659 | 86.600 | | | |
| 11 | 0.465 | 2.448 | 89.048 | | | |
| 12 | 0.434 | 2.284 | 91.332 | | | |
| 13 | 0.352 | 1.852 | 93.184 | | | |
| 14 | 0.318 | 1.676 | 94.861 | | | |
| 15 | 0.281 | 1.477 | 96.338 | | | |
| 16 | 0.231 | 1.214 | 97.552 | | | |
| 17 | 0.190 | 1.001 | 98.553 | | | |
| 18 | 0.144 | 0.757 | 99.310 | | | |
| 19 | 0.131 | 0.690 | 100.000 | | | |

From the analysis of Table 5, the first main factor (absolute value perspective) has a greater impact on the fifteen indexes of PR1, PR2, MR1, MR2, PV1, PV2, PV3, SN1, SN2, RP1, RP2, RP3, BC1, BC2, and BC3, which can be summarized as the perception factor; the second main factor (absolute value perspective) has a greater impact on the four indexes of PS1, PS2, PS3, and PS4, which can be summarized as the benefit-linking factor.

We used the statistical software Smart PLS (V 4.0.8) to test the reliability and validity of the research model using the confirmatory factor analysis method. The calculations of standard factor loading, average variance extracted (AVE), comprehensive reliability (CR), and Cronbach's alpha ($\alpha$) are shown in Table 6.

We examined data validity by inspecting each load of a single item. For each construct variable, all values were well over 0.50 for standard factor loading and were over 0.70 for CR. Except for the subjective norm, the value of Cronbach's alpha ($\alpha$) was greater than 0.70, which might be caused by insufficient sample size and observation variables. Values over 0.50 for AVE, of which the square root was greater than the value of related coefficients

between construct variables, indicated that the comprehensive reliability of the survey data were reliable and met the requirements [48]. Table 6 shows that the square root of the AVE value in the last row is greater than the correlation coefficient between dimensions in this study, indicating that the discrimination validity of each latent variable is better than the AVE values of each latent variable.

**Table 5.** Rotated component matrix.

|  |  | Component | |
|---|---|---|---|
|  |  | **1** | **2** |
| 1 | PS1 | 0.362 | 0.801 |
| 2 | PS2 | 0.287 | 0.786 |
| 3 | PS3 | 0.230 | 0.796 |
| 4 | PS4 | 0.078 | 0.565 |
| 5 | PR1 | −0.566 | −0.535 |
| 6 | PR2 | −0.645 | −0.472 |
| 7 | MR1 | −0.595 | −0.506 |
| 8 | MR2 | −0.579 | −0.290 |
| 9 | PV1 | 0.750 | 0.412 |
| 10 | PV2 | 0.760 | 0.471 |
| 11 | PV3 | 0.696 | 0.535 |
| 12 | SN1 | −0.573 | −0.199 |
| 13 | SN2 | −0.716 | −0.088 |
| 14 | RP1 | 0.623 | 0.340 |
| 15 | RP2 | 0.524 | 0.398 |
| 16 | RP3 | 0.547 | 0.340 |
| 17 | BC1 | 0.714 | 0.242 |
| 18 | BC2 | 0.697 | 0.205 |
| 19 | BC3 | 0.790 | 0.097 |

**Table 6.** Reliability and validity test (*N* = 264) of survey data.

| Construct | Items | Standardized Factor Loading | Related Coefficient * | Average Variance Extracted (AVE) | Composite Reliability (CR) | Cronbach's Alpha (α) |
|---|---|---|---|---|---|---|
| Benefits-linking stability (PS) | 4 | 0.547–0.901 | 0.460–0.681 | 0.648 | 0.877 | 0.810 |
| Perceived product risk (MR) | 2 | 0.893–0.897 | 0.526–0.737 | 0.773 | 0.872 | 0.711 |
| Perceived market risk (PR) | 2 | 0.845–0.912 | 0.597–0.826 | 0.801 | 0.889 | 0.751 |
| Perceived value (PV) | 3 | 0.899–0.929 | 0.606–0.810 | 0.850 | 0.944 | 0.912 |
| Risk Preference (RP) | 3 | 0.784–0.833 | 0.530–0.777 | 0.647 | 0.846 | 0.727 |
| Subjective norm (SN) | 2 | 0.820–0.881 | 0.460–0.657 | 0.724 | 0.840 | 0.622 |
| Green production behavior continuity (BC) | 3 | 0.806–0.834 | 0.532–0.810 | 0.674 | 0.861 | 0.758 |

Note: * Related Coefficient was linear correlation (R) between construct variables for explain discriminant validity. Source: Own calculations based on Smart PLS test results.

## 4. Results and Discussion

### 4.1. Fit Indices' Verification

We used the Path analysis and PLSpredict of Smart PLS (V 4.0.8) to evaluate the Fit indices of PLS-SEM based on the research model (in Table 7). For each construct variable, with all values over 0.33 for R-Square ($R^2$), values over 0.15 for Q-Square ($Q^2$), and values over 0.21 for the Goodness-of-Fit Index (GFI), the overall value was 0.063 below 0.08 for the Standardized Root Mean Square Residual error (SRMR). The overall value was 0.531 over 0.21 for GFI, indicating that the research model based on PLS-SEM has a good fit and the relevant results were acceptable [49].

### 4.2. Path Coefficients Estimation

We constructed a PLS-SEM to calculate the research hypothesis path. We estimated the path coefficients of the research model represented in Figure 1 using the path analysis

of Smart PLS (V 4.0.8) and built a variables path diagram (in Figure 2) to represent the hypothesized relationships linking the constructs.

**Table 7.** Fit indices of PLS-SEM on the research model.

| Fit Indices | Model Value | Reference Value |
|---|---|---|
| Standardized Root Mean Square Residual error (SRMR) | 0.063 | <0.08 |
| Overall Goodness of Fit Index (GFI) * | 0.531 | >0.21 |
| R-Square ($R^2$) | 0.357–0.804 | >0.33 |
| Q-Square ($Q^2$) | 0.347–0.663 | >0.15 |
| Root Mean Square of Error (RMSE) | 0.585–0.815 | Smaller |
| Mean Absolute Error (MEA) | 0.430–0.622 | Smaller |
| Goodness of Fit Index (GFI) * | 0.352–0.707 | >0.21 |

Note: * GFI = $\sqrt{R^2 \times Q^2}$; $N = 264$. Source: Own calculations based on Smart PLS test results.

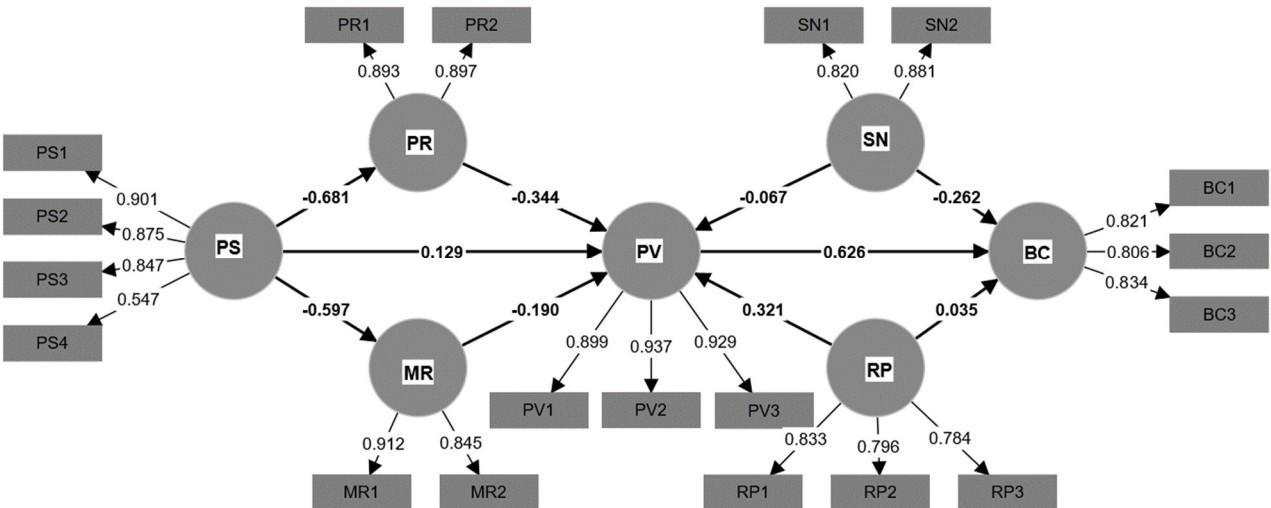

**Figure 2.** Path diagram with path coefficient to estimate research model. Source: Own calculations based on Smart PLS test results.

Following the path diagram in Figure 2, the hypothetical test results for the path coefficient values (β, standardized in a range from −1 to +1), standard deviation (STDEV), and P values were as shown in Table 4, in which we found that a total of eight assumptions were significantly accepted and in accordance with our expectations.

As shown in Table 8, the correlation coefficients (r) indicated that the benefit linkage related to the perceived product risk (MR), perceived market risk (PR), perceived value (PV) (r = −0.597, −0.681,0.805), and perceived risk (MR,PR) was highly related to the perceived value (r = −0.737, −0.826). The perceived value was related to green production behavior continuity (BC) (r = 0.810). The subjective norm (SN) was related to perceived value and BC (r = −0.602, −0.657), while risk preference (RP) was related to perceived value and BC (r = 0.660, 0.777).

Moreover, as shown in Table 4, the results showed that the stable benefit linkage had a significantly negative affect on the production risk perception (path coefficient or direct relation β = −0.681, *p* < 0.001) and market risk perception (β = −0.597, *p* < 0.001) and positively affected the perceived value (β = 0.129, *p* < 0.050), so, the hypotheses (H1a-, H1b- and H2+) were accepted. Production risk perception (β = −0.344, *p* < 0.001) and market risk perception (β = −0.190, *p* < 0.001) had a significantly negative affect on the value perception, so the hypothesis (H3a- and H3b-) was accepted. The perceived value had a significantly positive effect on the farmers' green production behavior continuity (β = 0.626, *p* < 0.001), so the hypothesis (H4+) was accepted. The subjective norm had a significant negative effect on the farmers' green production behavior continuity (β = −0.262, *p* < 0.001), so the

hypothesis (H5b-) was supported. Additionally, the risk preference had a significantly positive effect on the perceived value (β = 0.321, *p* < 0.001), so, the hypothesis (H6b+) was supported.

**Table 8.** Path coefficients estimation of research model to hypothesized test.

| Hypothesis | Path | Standard Deviation (STDEV) | T Statistics (∣β/STDEV∣) | Path Coefficient (β) | Tested Result |
|---|---|---|---|---|---|
| H1a- | PS- > PR | 0.045 | 15.007 | −0.681 *** | Accepted |
| H1b- | PS- > MR | 0.050 | 12.026 | −0.597 *** | Accepted |
| H2+ | PS- > PV | 0.053 | 2.462 | 0.129 ** | Accepted |
| H3a- | PR- > PV | 0.060 | 5.721 | −0.344 *** | Accepted |
| H3b- | MR- > PV | 0.042 | 4.504 | −0.190 *** | Accepted |
| H4+ | PV- > BC | 0.059 | 10.627 | 0.626 *** | Accepted |
| **H5a-** | **SN- > PV** | **0.041** | **1.629** | **−0.067 n.s.** | **Rejected** |
| H5b- | SN- > BC | 0.055 | 4.726 | −0.262 *** | Accepted |
| H6a+ | RP- > PV | 0.050 | 6.363 | 0.321 *** | Accepted |
| **H6b+** | **RP- > BC** | **0.059** | **0.591** | **0.035 n.s.** | **Rejected** |

Note: ** *p* < 0.050, *** *p* < 0.001, n.s. not significant; T-statistic > 1.645 (*p* = 0.05); *N* = 264. Source: Own calculations based on Smart PLS test results.

### 4.3. Path Effects Calculations

In order to test the path mediation effects of the perceived product risk (MR), perceived market risk (PR), and perceived value (PV) on the hypothesized model, the bootstrapping analyses function of Smart PLS (V 4.0.8) was applied to conduct 5000 repeated sampling tests on these variables. A regression analysis using SPSS (V13.0) was performed to test the moderation effects of the risk preference (RP) and subjective norm (SN) on the research model. The estimated results of path mediation and moderation effect are shown in Table 9.

**Table 9.** Path mediation and moderation effects estimation of research model.

| Path | Standard Deviation (STDEV) | T Statistics (∣β/STDEV∣) | Path EFFECTS (β) | *p* Values |
|---|---|---|---|---|
| **Path Mediation Effects** | | | | |
| MR- > PV- > BC | 0.031 | 3.820 | −0.119 *** | 0.000 |
| PR- > PV- > BC | 0.036 | 5.943 | −0.215 *** | 0.000 |
| PS- > BC | 0.047 | 6.330 | 0.298 *** | 0.000 |
| PS- > PV | 0.048 | 7.221 | 0.348 *** | 0.000 |
| RP- > PV- > BC | 0.039 | 5.202 | 0.201 *** | 0.000 |
| SN- > PV- > BC | 0.027 | 1.550 | −0.042 n.s. | 0.121 |
| **Path moderation effects** | | | | |
| RP × PV- > BC | 0.028 | 2.646 | 0.074 *** | 0.000 |
| SN × PV- > BC | 0.023 | 4.427 | −0.103 ** | 0.008 |

Note: ** *p* < 0.050, *** *p* < 0.001, n.s. not significant; T-statistic > 1.645 (*p* = 0.05); *N* = 264. Source: Own calculations based on Smart PLS and SPSS test results.

As shown in Table 9 and Figure 2, the mediation effects (direct relation, β) of path (MR->PV->BC) and path (PR->PV-> BC) were −0.119 and −0.215 (*p* < 0.001), which indicated that the perceived product risk (MR) and perceived market risk (PR) had a significant negative effect on green production continuity (BC) through the mediated perceived value (PV), and the effect of the production process was greater than that of market operation. The total mediation effect of path (PS -> BC) was β = 0.298 (*p* < 0.001), based on path (PS ->MR -> PV -> BC), path (PS -> PR -> PV -> BC), and path (PS -> PV -> BC) which showed that the benefits-linking stability (PS) had a significantly positive effect on BC.

In addition, the mediating effect of path (RP -> PV -> BC) was β = 0.201 (*p* < 0.001), with a moderation effect β = 0.074 (*p* < 0.001) of path (RP × PV -> BC), which indicated that the risk preference (RP) had a significant positive effect on BC by mediating PV and had a significant positive moderating role between PV and BC. The moderation effect β = 0.103

($p < 0.05$) of path (SN × PV -> BC), showed that the subjective norm (SN) negatively moderated the effect between PV and BC significantly. Moreover, in Figure 3, the influence of the perception value (PV) on green behavior continuity (BC), high subjective norms, and high-risk preference had a greater moderation effect than the low subjective norms and low-risk preference achieved because of the linear slope $K_{high} > K_{low}$.

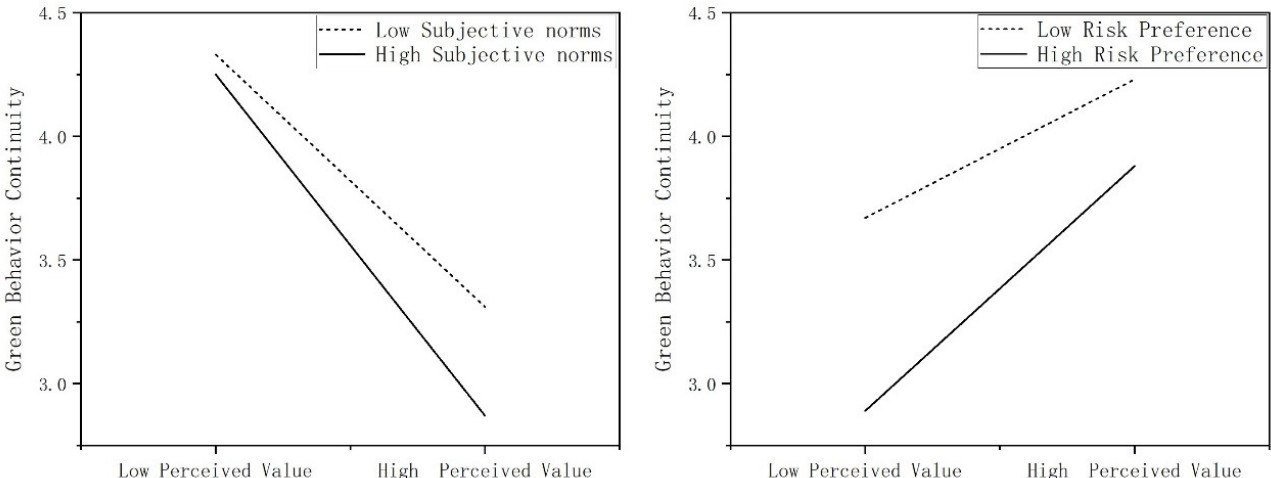

**Figure 3.** The moderating effect of subject norms (SN) and risk preference (RP) on the relation between perceived value and green production continuity (BC).

*4.4. Discussion*

The research findings provide a theoretical and practical value for farmers, allowing them to tackle the risks of COVID-19 and improve their green production based on platform organization in C2F.

First, the risks of the COVID-19 pandemic significantly impacted the green production continuity of family farms, which reduced green technology adoption intentions (BC1, load = 0.821), increased the cost of living (BC2, load = 0.806), and cut down moderate-scale effects (load = 0.834). The study provided evidence of how risk perception affects farmers' behavior through a mediated perception value. The series of pandemic prevention measures implemented in response to the COVID-19 pandemic, such as the lockdown and quarantine policies [1,50], caused difficulties in obtaining materials and prevented farmers from going out and doing their jobs. The traffic control measures of COVID-19 caused blocked logistics and violent fluctuations in the market prices of products. For moderate-scale family farmers, the shortage of capital placed more pressure on production (load = 0.897) than that of materials and labor (load = 0.893), resulting in a huge perceived production risk (PR), while the impact of sales channel blockage (load = 0.912) on product sales was more direct. This factor aggravates perceived market risk (MR) [4,5]. Perceived risk negatively affected the perception value at risk [37] measured by technical reliability (load = 0.899), service satisfaction (load = 0.937), and platform loyalties (load = 0.929), which in turn negatively affected farmers' green production decisions and behaviors continuity (BC).

Additionally, the stable benefit linkage of a platform organization in the C2F model could significantly reduce the risk of COVID-19 for the green production continuity of family farms. This research offered and demonstrated a new perspective and approach for farmers' green production to tackle COVID-19. We found that, among the measuring factors of benefits-linking stability (BC) in C2F, platform value cognition (PV1) had the highest load (0.901), followed by the participation co-operation (PS2, 0.875), access and benefit distribution (PS3, 0.847), and finally, safeguard and restraint measures (PS4, 0.574). These showed that greater participation in platform organization in C2F, especially with the aim of enhancing awareness of benefit-linking and participation in benefit distribution, could strengthen the stability and compactness of benefit-linking on platform organization

in C2F and allow farmers to obtain more green technology and market information [9,10]. This will help to effectively alleviate the uncertainty of production and market caused by the COVID-19 pandemic, reduce the perception of risk, and increase perception value and improve family farms' green production continuity.

Moreover, risk preference and subjective norms had moderated effects on the farmers' green production continuity. We explained the behavior motivation of farmers with different risk preferences in response to the COVID-19 pandemic. The subjective norms and risk preferences of farmers affected the farmers and influenced their green behavior willingness. The risk preference measured as risk aversion (load = 0.833), risk neutral (load = 0.796), and risk appetite (0.784) had a significant positive effect on the risk value perception, which promoted green production [22]. When the degree of farmers' preference was higher, those who perceived a greater risk value were more willing to adopt green production behaviors, while farmers with a lower risk perception were unwilling to adopt green production. Compared with the organization impact (load = 0.820), policies (load = 0.881) had a greater impact on farmers' production behavior. Unlike traditional subjective norms, such as ecological protection policies that positively affect the farmers' green production behavior, this research mainly tested policy in relation to the community impact of the COVID-19 pandemic, which had a significant negative impact. When farmers are greatly affected by subjective norms, their willingness to adopt green production with a greater perceived risk value will decline at a more significant rate.

## 5. Conclusions and Suggestions

### 5.1. Conclusions

Based on the C2F we developed for family farmers' green products, regional family farmers (users) were taken as the research objects. Applying risk decision theory, we built a theoretical analysis framework based on platform organization, pandemic risk, perception value, risk preference, subjective norms, and farmers' willingness, and ascertained the impact effects and path using the PLS-SEM model. We analyzed the impact of the risk of the COVID-19 pandemic on farmers' green production willingness, demonstrated the potential of the digital platform organization in terms of resistance to pandemic risk, and studied the regulation effects of subjective norms and risk preference on the digital platform organization in relation to farmers' resistance to pandemic risk. Finally, we proposed suggestions and measures for maintaining farmers' livelihoods through the sale of green products to tackle COVID-19. The results showed that the risks of the COVID-19 pandemic significantly affected family farms' green production continuity, which reduced green technology adoption intentions, increased livelihood costs, and cut down moderate-scale effects. However, the stable benefit linkage of platform organizations based on the C2F model could significantly reduce the risk of COVID-19 for family farms' green production continuity. We also explained the behavior motivation of farmers with different risk preferences in response to the COVID-19 pandemic. The risk preference and subjective norms had visible effects on farmers' green production continuity.

### 5.2. Suggestions

**Perfect the functions of digital platform organizations to optimize the benefit linkages in C2F**. The willingness of farmers to adopt green production increased with the improvement in the degree of interest linkage with C2F platform organizations. We suggest strengthening the organization and construction of the C2F platform, which will improve the mechanism of benefit-linking. Due to the COVID-19 pandemic, business entities have been seriously affected by traffic and personnel flow control, so the advantage of this platform is that it can break the spatial barrier and provide more agricultural production technology training for farmers online. Holding a variety of production exchanges and other activities to publicize and clarify the purpose, content, and significance of the activities of farmers can mobilize farmers' enthusiasm to participate in collective activities,

promote their communication and interaction, and enhance the level of trust between farmers and C2F platform organizations.

**Accelerate farmers' digitization abilities for cultivation to increase their cognitive risk level**. Strengthen the guidance and training for the digital ability of farmers. Through education and training, farmers' knowledge can be expanded, and their cognitive ability, understanding ability, and acceptance of digital platforms can be improved. Farmers' awareness and penetration rate of digital platforms and technologies can be promoted. Then, the large data from the platform can be used to strengthen their grasp of the market and their resource integration ability during the COVID-19 pandemic, including the distribution of green crops, market information on green agricultural products, the collection of market supply and demand, agricultural prices, and other big data to improve their risk awareness. Production and marketing docking can be strengthened, and sales channels expanded. Through digital platforms, farmers can effectively capture the wave of community "group buying" caused by the isolation of community residents during the pandemic and alleviate the market risk.

**Strengthen the policy guidance of COVID-19 prevention to reduce the influence of farmer subjective norms**. We should take full advantage of the moderating role of subjective norms. Local agricultural upgrading is influenced by local forces [51]. On the one hand, the government can encourage farmers to participate in green production through subsidies and awards to ease the economic pressure on farmers or strengthen policy guidance. For example, during the COVID-19 pandemic control period, agricultural products such as vegetables, livestock, and poultry should be included in the "green channel" for emergency transportation so as to ensure prioritization and convenient access to ease logistics and sales congestion. On the other hand, in rural societies, loudspeakers, the internet, TV, newspapers, posters, publicity columns, and other resources can be used to strengthen the publicity of green production so that the development of green ecological agriculture becomes a social consensus. Then, we can positively adjust farmers' willingness to carry out green production.

**Author Contributions:** Writing—original draft, L.M.; Writing—review & editing, J.S.; Data curation and Investigation, S.X.; Supervision and corresponding authors, D.Y. All authors have read and agreed to the published version of the manuscript.

**Funding:** Special thanks to the Jiangsu University Philosophy and Social Science Research Major Project (2022SJZD091).

**Institutional Review Board Statement:** Not applicable.

**Informed Consent Statement:** Informed consent was obtained from all subjects involved in the study.

**Data Availability Statement:** The data used to support the findings of this study are cited in relevant places within the text as references.

**Conflicts of Interest:** The authors declare no conflict of interest.

## Appendix A. A Measurement Variables and Observed Items

**Table A1.** Measurement variables of the farmers' organized behavior to treat the risk of COVID-19 on C2F.

| Research Variables | Measurement Items (Observed Variables) |
| --- | --- |
| PS1-Platform value Cognition | $PS_{01}$: Will learn more production techniques from the C2F platform organization. <br> $PS_{02}$: Will get more services from the C2F platform organization. |
| PS2-Participation Co-Operate | $PS_{03}$: Will learn to adopt green agricultural technology in time. <br> $PS_{04}$: Will actively exchange green agricultural technology with other farmers. |

**Table A1.** *Cont.*

| Research Variables | Measurement Items (Observed Variables) |
|---|---|
| PS3-Access & benefit distribution | $PS_{05}$: Will improve the production of agricultural products by participating in the C2F platform organizations. |
| | $PS_{06}$: Will get more Revenue by participating in the C2F platform organizations. |
| PS4-Safeguard & restraint measures | $PS_{07}$: The management of the C2F platform organization is satisfactory. |
| | $PS_{08}$: Rights are guaranteed in the C2F platform organization. |
| PR1-Materials & Labor shortage | $PR_{01}$: There is less labor involved in agricultural production. |
| | $PR_{02}$: It Is difficult to buy agricultural production materials. |
| PR2-Financial pressure | $PR_{03}$: No more money to put into production. |
| | $PR_{04}$: Face the threat of bankruptcy by borrowing capital. |
| MR1-Sales channel blocked | $MR_{01}$: Difficult to sell agricultural products amid the COVID-19 pandemic. |
| | $MR_{02}$: Difficult to transport produce to market. |
| MR2-Market price volatility | $MR_{03}$: Market price fluctuations lead to a decline in economic benefits. |
| | $MR_{04}$: Green produce is harder to sell at a good price in the market. |
| PV1-Technical reliability | $PV_{01}$: Satisfaction of farm using RFS to increase productivity, reduce cost and risk. |
| | $PV_{02}$: Easily understand and learn function system of RFS on green digital platform. |
| PV2-Service satisfaction | $PV_{03}$: Easily use and operate RFS in farm green production under the servive of C2F platform. |
| | $PV_{04}$: Easily obtain and use digital service for using RFS in farm management. |
| PV3-Platform loyalties | $PV_{05}$: Will continue to use the Green Agriculture Platform in the future |
| | $PV_{06}$: The C2F Platform will help solve the difficulties when there are technical risks |
| SN1-Community impact (SN1) | $SN_{01}$: Grow the same produce as everyone else. |
| | $SN_{02}$: Join the platform organization with neighbors, related relatives, friends, colleagues and other members of society. |
| SN2-Government influence | $SN_{03}$: The quarantine policy has reduced agricultural production opportunities. |
| | $SN_{04}$: Agricultural subsidies increase willingness to produce. |
| RP1-Risk aversion | $RP_{01}$: Farm insurance is not important, expand production and generate income through loans. |
| RP2-Risk neutral | $RP_{02}$: The greater the risk, the greater the benefit. |
| RP3-Risk appetite | $RP_{03}$: Expand production and generate income through loans. |
| BC1-Green Product effects | $BC_{01}$: Effect of pesticides-fertilizer-water reduction by using green production technology in production. |
| | $BC_{02}$: Green technology adoption intention. |
| BC2-Economic effects | $BC_{03}$: Effect of labor reduction and production creasing by using green production technology in production. |
| | $BC_{04}$: Effect of saving cost by using green production technology in production. |
| BC3-Scale extension effects | $BC_{05}$: Effect of recommending the green production technology to other farmers adopt. |
| | $BC_{06}$: Effect of expand agricultural production. |

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
