# Peer review of "Impact of Digital Platform Organization on Reducing Green Production Risk to Tackle COVID-19: Evidence from Farmers in Jiangsu China"

_agriculture, doi:10.3390/agriculture13010188_

Round 1
Reviewer 1 Report
This is a topical area of interest with the potential to attract readership. Some general comments are as follows:
1. Introduction can be strengthened with some details on the types of uncertainties that were introduced during the pandemic (see line 40), a discussion on the extent to which the digital platform has infiltrated farming practices in China, and an indication as to why social capital, though mentioned, was not included within the study.
2. Reliability Score for the subjective norm measure was 0.622. Was factor analysis employed to explore the possibility of improving this measure? if not, what are the limitations of the findings based on the use of this measure, which stands at less than the require 0.7 standard.
3. Discussion-These are quite interesting findings that require some comparison to the existing literature. Key questions therefore are for the extent to which these findings are consistent or not with existing research? The implications for further research and for the relevance of the theory used in the study. These require more in-depth discussion.
Author Response
Response to Reviewer 1 Comments
Point 1: Introduction can be strengthened with some details on the types of uncertainties that were introduced during the pandemic (see line 40), a discussion on the extent to which the digital platform has infiltrated farming practices in China, and an indication as to why social capital, though mentioned, was not included within the study.
Response 1: In the article, I had strengthened the discussion on the uncertainty of agricultural production during the COVID-19 epidemic. The deepening of the social division of labor system and the long period of agricultural investment return have caused the uncertainty and added relevant explanations
要点 2:主观规范测量的可靠性得分为0.622。是否采用了因素分析来探索改进这一措施的可能性?如果不是,基于该措施的使用,调查结果的局限性是什么,该措施低于要求0.7标准。
响应 2:相关解释已添加到文章中。主观范数测量的可靠性得分为0.622可能是由样本量和观测变量不足引起的。
要点3:讨论 - 这些都是非常有趣的发现,需要与现有文献进行比较。因此,关键问题是这些发现在多大程度上与现有研究一致?对进一步研究和研究中使用的理论的相关性的影响。这些都需要更深入的讨论。
响应 3:根据您的建议,我在讨论部分添加了与现有文献的比较,以增强讨论的深度

Reviewer 2 Report
1. This paper has merits, but they will be realized only in a substantial revision.
2. There are sentences without verbs here and there. In the abstract, for instantce. I suggest that the authors read the paper aloud to themselves in a quiet room. They will hear the awkward or inappropriate phrasings. Or hire an English editor.
3. Long paragraphs lose the reader in verbiage. Make 3-5 sentence paragraphs that focus on an idea with appropriate transitions.
4. Methods are problematic. Data are supposedly a random sample, but no specifics of how the respondents were selected, contacted, or the actual response rate. This is scientifically vital to clarify.
5. Measurement requires clarification. Just what was asked, how is it coded. Every variable needs a clear explanation.
6. The factor analysis is not clearly explained. What variables were analyzed, factor loadings, eignevalues, variance explained. It is not clear how the reliablity coefficients presented were computed or what they mean. Are seven decimal points necessary? The authors should consult examples in the published literature. Find highly cited presentations using principle components factor analysis for examples to emulate.
7. As the methods and analysis is problematic, it does not seem productive to consider the conclusion until the above matters are resolved.
Author Response
Response to Reviewer 2 Comments
Point 1: Methods are problematic. Data are supposedly a random sample, but no specifics of how the respondents were selected, contacted, or the actual response rate. This is scientifically vital to clarify..
Response 1: I have explained the specific research in the data source and explained how to contact the respondents:Face-to-face interviews with farmers while we assist them in filling out question-naires。
Point 2: Measurement requires clarification. Just what was asked, how is it coded. Every variable needs a clear explanation.
Response 2: The variable measurement method is further explained. See 3.1(line202) for details
Point 3: The factor analysis is not clearly explained. What variables were analyzed, factor loadings, eignevalues, variance explained. It is not clear how the reliablity coefficients presented were computed or what they mean. Are seven decimal points necessary? The authors should consult examples in the published literature. Find highly cited presentations using principle components factor analysis for examples to emulate.
Response 3: Following your suggestion, I have added factor loadings, eignevalues, variance explained in the paper.
Round 2
Reviewer 2 Report
Saying that the respondents were chosen via random sampling is not enough.
What lists were used? What process was used to select individuals? How were refusals handled? Who was chosen to speak for the farm?
The conclusion to be drawn from the olim test and what it means should be clarified.
Means, factor coreffiecients, and regression coefficients should have no more than three decimals points. How precise is your data?
Author Response
Response to Reviewer 2 Comments
Point 1: Saying that the respondents were chosen via random sampling is not enough. What lists were used? What process was used to select individuals? How were refusals handled? Who was chosen to speak for the farm?
Response 1: I had further explained the sample selection and research process in the article(see line 235).
Point 2: Means, factor coreffiecients, and regression coefficients should have no more than three decimals points. How precise is your data?.
Response 2: All the data were checked to retain three decimal places.
Response 3: I have finished the English expression specification
